# Nanoparticle-Induced Property Changes in Nematic Liquid Crystals

**DOI:** 10.3390/nano12030341

**Published:** 2022-01-21

**Authors:** Nicolas Brouckaert, Nina Podoliak, Tetiana Orlova, Denitsa Bankova, Angela F. De Fazio, Antonios G. Kanaras, Ondrej Hovorka, Giampaolo D’Alessandro, Malgosia Kaczmarek

**Affiliations:** 1School of Physics and Astronomy, University of Southampton, Southampton S017 1BJ, UK; N.Podoliak@soton.ac.uk (N.P.); d.o.bankova@soton.ac.uk (D.B.); a.kanaras@soton.ac.uk (A.G.K.); 2Istituto Italiano di Tecnologia, Via Morego 30, 16163 Genova, Italy; angela.defazio@iit.it; 3Faculty of Engineering and Physical Sciences, University of Southampton, Southampton S017 7QF, UK; o.hovorka@soton.ac.uk; 4School of Mathematics, University of Southampton, Southampton S017 1BJ, UK

**Keywords:** liquid crystal, nanoparticles, colloidal suspension, characterisation, optical multi-parameter analyser, elastic constants, rotational viscosity

## Abstract

Doping liquid crystals with nanoparticles is a widely accepted method to enhance liquid crystal’s intrinsic properties. In this study, a quick and reliable method to characterise such colloidal suspensions using an optical multi-parameter analyser, a cross-polarised intensity measurement-based device, is presented. Suspensions characterised in this work are either plasmonic (azo-thiol gold AzoGNPs) or ferroelectric Sn_2_P_2_S_6_ (SPS) nanoparticles in nematic liquid crystals. The elastic constants and rotational viscosity showed nonlinear dependence on the concentration of AzoGNPs, initially increasing at lower concentrations and then decreasing at higher concentrations, indicating some degree of particle aggregation. For the SPS suspension, the elastic constant decreased with doping, while the rotational viscosity increased, in agreement with previous findings. Through viscosity measurements, the stability of SPS suspension over ten years is also highlighted.

## 1. Introduction

The electro-optic properties of liquid crystals (LC) are widely exploited in optical devices, such as displays, light modulators and variable wave plates [1,2,3]. Their birefringence and dielectric anisotropy, coupled with their elastic properties, allow these materials to be quickly and reversibly addressed by an external electric or magnetic field, making liquid crystals perfect for commercial displays. However, as the general technology evolves, requirements for the liquid crystals change too. A shorter response time or lower operating voltage are common examples of LC characteristics that must be continuously improved to match the increasing demand.

A solution to avoid chemical synthesis of new types of LCs is to dope existing LCs with nanoparticles to change their properties [1,2,3,4,5,6,7,8,9]. A variety of nanoparticles such as ferroelectrics [10,11,12,13,14,15], gold [5,6,16,17,18,19,20,21], quantum dots [22,23,24,25,26,27,28], oxides [4,7,29] and magnetic nanoparticles [30,31,32] have been tested in LCs. It has been reported that doping nematic liquid crystals (NLC) with ferroelectric nanoparticles such as Sn_2_P_2_S_6_ (SPS) or BaTiO_3_ (BTO) can efficiently decrease the driving voltage, enhance the dielectric response [1,10,11] and increase the optical anisotropy [14]. The main mechanism behind such effects is the coupling between the natural polarisation of the nanoparticles and the LCs, increasing the molecular order and the electro-optical properties of the blend [15]. The impact of the ferroelectric nanoparticles highly depends on their concentration and size. The concentration is usually kept below 1 wt% to avoid significant particle aggregation, while the size is in the range of 10 to 100 nanometres [11,12]. Smaller particles may lose their ferroelectric properties, while bigger ones can contain multiple ferroelectric domains [11,13].

Another type of nanoparticle, widely employed with LCs, are metallic nanoparticles, such as gold. These improve the electro-optical properties of LCs by, once again, increasing the molecular order of the blend [1,5,6,33]. Gold nanoparticles also possess plasmonic resonances, leading to an increase in the nonlinear optical properties of the LC [6,17,18,19]. Another advantage of gold nanoparticles (GNPs) is the possibility of functionalising them with different types of surfactants, including mesogenic ligands. Functionalising gold nanoparticles increases their solubility in the LC host, making it possible to obtain stable suspensions with a higher concentration [5,21]. It has also been demonstrated that functionalising GNPs with azo-thiol ligands can also induce a reversible change of alignment of the 5CB LC upon light irradiation, from homeotropic under UV to planar when irradiated by visible light [20].

However, doping with NPs may also lead to particle aggregation. Its impact on the LC can be either beneficial, by reducing its viscosity by generating clusters that leave more space for the LCs to move freely [6,10], or detrimental, by degrading its elastic properties [8] or causing light scattering. These contrasting effects must be well understood and controlled to ensure that the suspension has the desired properties.

With the expanding research on doped LCs, there is an increasing need for a quick and reliable method of quantifying the electro-optical and physical properties of LC composites. In this study, we propose a method to characterise colloidal suspensions using an optical multi-parameter analyser (OMPA), a method originally designed for undoped LCs [34,35]. It measures critical LC properties such as elastics constants (splay elastic constant K_1_ and bend elastic constant K_3_), pretilt θ, anchoring energy W_p_ and rotational viscosity γ_1_ from simple cross-polarised intensity experiments (CPI). We demonstrate, for the first time, the use of OMPA to study the changes of E7 and 5CB electro-optical and viscosity properties when doped with novel nanoparticles, azo-thiol-functionalized gold nanoparticles (AzoGNPs), with the concentrations up to 3 wt%. We also measure the properties of nematic LC18523 doped with 1 wt% SPS nanoparticles and test the stability over time. We compare its elastic constants and rotational viscosity from previously published results [35] and extend its characterisation by studying the impact of ageing on these cells on both elastic constants and rotational viscosity. We, therefore, demonstrate the ability of the OMPA technique to easily characterise the key properties of both pure and doped LCs.

## 2. Materials and Methods

### 2.1. Particles Synthesis

The synthesis of small gold nanoparticles functionalized with Azo-C11-SH ligands was achieved following previously published protocols [20,36]. Briefly, an aqueous solution of HAuCl_4_ (1.5 mL, 30 mM) was mixed with a solution of tetraoctylammonium bromide (TOAB) in toluene (4 mL, 50 mM). The two-phase mixture was rapidly stirred until all Au^3+^ was transferred to the organic phase to produce a deep-red solution. After discarding the aqueous phase, Azo-C11-SH (26 µmol) was added to the solution (Gold:Azo-C11-SH ≈ 2:1). Then, NaBH_4_ (1.5 mL, 0.3 M) was slowly added under vigorous stirring. The reaction stirred for 2 h, and the product was purified by centrifugation using ethanol (3×, 5000 rpm). The Au core of nanoparticles was 5 nm in diameter, as measured by transmission electron microscopy (TEM). Finally, the Azo–gold nanoparticles were stored in toluene (concentration of 2.5 mg/mL) at 4 °C until further use.

HAuCl_4_, TOAB, NaBH_4_ and all the solvents were purchased from Merck Sigma-Aldrich (Burlington, MA, USA) and used without further purification. The Azo-C11-SH ligand was purchased from Prochimia Surfaces (Gdynia, Poland).

Sn_2_P_2_S_6_ (SPS) nanoparticles were prepared by milling micron-sized particles in a vibration mill. Surfactant (oleic acid) and a solvent (heptane) were added during the milling to stabilize the particle suspension. The resulting average size of nanoparticles was 50 nm, as confirmed by atomic force microscopy (AFM) and TEM [35,37].

### 2.2. Suspension Preparation

Suspensions of AzoGNPs in nematic LCs E7 and 5CB, with a weight content of 1 wt% in E7 and 5CB and 3 wt% in 5CB, were prepared by the following method. A total of 0.4 mL and 1.2 mL of AzoGNPs in toluene was added to 0.1 g of liquid crystals for 1 wt% and 3 wt% concentrations, respectively. The mixture was sonicated for several minutes to obtain a uniform blend, heated to 60 °C, and left overnight under an extraction hood to evaporate the solvent. The same technique was used to prepare a suspension of 1wt% SPS nanoparticles in LC18523.

### 2.3. Cell Preparation

All measurements were performed using planar liquid crystal cells. The cells consisted of a thin layer of pure/undoped LC or nanoparticle suspensions between two glass slides. In the case of pure and doped LC18523, the cells studied in this paper were prepared ten years ago by our group. The gap between glass slides was fixed using 12 µm spacers. The inner surfaces of the glass slides were covered with a conductive ITO layer and a rubbed polyimide alignment layer to obtain a planar orientation of the liquid crystal inside the cells. The LC18523 cells were stored in air at constant room temperature; all new measurements in this study were performed on the cells without any reheating, remixing or sonication. For the pure and doped E7 and 5CB cells, commercial planar cells purchased from EHC Co., Ltd. (Hachioji-shi, Tokyo) were used, with an ITO-coated surface of 50 mm^2^, polyimide alignment layers and a set gap between the glass slides of 10 ± 0.5 µm.

All these cells were filled with either undoped LCs or suspensions in an isotropic phase using capillary forces and sealed using an epoxy glue to avoid leakage and contamination of the LC over time. Good-optical-quality liquid crystal alignment and suspension homogeneity across the cell were obtained. All measurements were performed at room temperature, approximately 22 °C, where the pure and doped LCs were in the nematic phase.

### 2.4. CPI and Viscosity Measurements

To measure and characterise our samples, we used a CPI setup [34] consisting of a planar liquid crystal cell placed between two crossed polarisers (polariser and analyser), with the LC director field at an angle of 45° with the axes of polarisers. The cell was illuminated by a laser with a beam area of about 7 mm^2^ that defined the studied area of the samples. Two photodiodes recorded the light intensity before the polariser and after the analyser. We used 642 nm and 532 nm fibre-coupled diode lasers as beam sources. The schematic drawing of the setup is presented in Figure 1.

This setup was controlled by a software, which performs data acquisition and analysis [34]. The cells were addressed with a sinusoidal AC signal at 10 kHz, the amplitude of which varied from 0 to 10 V with a voltage step of 0.02 V. The average value of the CPI was measured at each voltage step.

OMPA built-in algorithms fit CPI traces to determine elastic constants, cell thickness and pretilt angles. However, the dielectric constants and refractive indices of the suspensions should be specified for the numerical model to run correctly. The refractive indices were obtained from the literature [14,35,38]. Dielectric constants were obtained by measuring the capacitance of LC cells before and after filling them with the suspension using an automatic precision bridge (B905) from Wayne Kerr. The parallel component of the dielectric tensor, **ε_//_**, was measured using a cell with homeotropic alignment, while the perpendicular component, **ε**_⊥_, was obtained using a planar cell.

Viscosity measurements were carried out using the same CPI setup, following a method reported by us previously [39]. The cell was addressed with a high-frequency (10 kHz)-fixed voltage that had a low-frequency amplitude modulation. The frequency of the amplitude modulation varied between 0.1 and 10 Hz. The driving voltage was selected to have a value above the Fredericks threshold, such that the CPI was around 50% of the maximum value. The modulation amplitude was selected so the CPI oscillated between 20% and 80%. We measured the standard deviation of the CPI oscillations, std(CPI), as a function of the frequency of the amplitude modulation. At low-amplitude modulation frequency, the CPI follows the variation of the driving signal amplitude. However, when the frequency of the amplitude modulation increases, the std(CPI) variation decreases as the LC is not able to follow the driving signal. OMPA fits the slope of the std(CPI) curve, evaluating the rotation viscosity of the LC suspension [39].

The OMPA set-up, in principle, can map a large area of a cell, determined by the size of the illuminating beam and the size of the ITO region [40]. However, in case of commercial cells, the limited size of the electrode did not allow us to probe the cells at different points across the cells.

## 3. Results and Discussion

In this section, we present OMPA measurements of elastic constants and rotational viscosity of nematic LCs E7 and 5CB doped with azo-thiolated gold nanoparticles and LC18523 doped with ferroelectric nanoparticles (SPS).

### 3.1. Elastic Constants of AzoGNPs-Doped Liquid Crystals

Azo-thiol gold nanoparticles (AzoGNPs) were synthesised by the method described in Section 2.1. We selected LCs that have been well studied in the literature as a base for the LC AzoGNPs suspensions, namely E7 and 5CB [9,38,41]. The suspensions with a particle content of 1 wt% AzoGNPs in both E7 and 5CB and 3 wt% AzoGNPs in 5CB were prepared. These amounts were chosen taking into account the relative stability of the NPs in the liquid crystal suspension, with the aim of having stable, well-dispersed NPs. Azo-GNP dispersed in nematic LCs have, so far, only been studied in 5CB, a single-component liquid crystal [20]. Here, we expanded this study to include Azo-GNP suspensions in E7 liquid crystal, a mixture of several mesogenic molecules. In addition, we investigated nematic LCs with ferroelectric nanoparticles, LC18523 liquid crystal doped with 1 wt% SPS. This suspension has already been explored in our previous work [35]. In the current study, we aimed to explore the stability of LC18523 + 1%SPS over time, testing the cells prepared more than ten years ago.

As a first step, we investigated how elastic constants and viscosity change in LCs doped with NPs. The results of experimental CPI curves, and their fitting obtained by OMPA for 5CB-based AzoGNPs suspensions, are shown in Figure 2. The measured characteristics of all suspensions are summarised in Table 1. We can see from Figure 2 that the fits obtained by the OMPA match the experimental data minima and maxima very well. This ensures a precise estimation of both K_1_ and K_3_ [40]. The fit for doped E7 is available in the Appendix A, also showing a good match of the minima and maxima with the experimental data. From these measurements, we can see that the OMPA is perfectly able to model stable LC–nanoparticle composites.

From Table 1, we noticed that the splay elastic constant K_1_ and dielectric permittivity Δε tended to decrease by 17% and 30% in E7 + 1%AzoGNPs and by 32% and 40% in 5CB + 3%AzoGNPs, respectively. As reported in the literature, the process of particle aggregation may cause this behaviour [42]. Nanoparticles tended to disturb LC orientation in their neighbourhood. In the case of GNPs, the LC molecules interacted with the ligands on the surface of GNPs that locally distort LC orientation. This local distortion of LC around nanoparticles was more prominent for larger particles or as particle aggregation occurred. The small-size nanoparticles, however, can be incorporated into the LC matrix with minimum distortion. We noticed that the distortion of LC ordering can be observed by a decrease in elastic constants, as well as in dielectric anisotropy with doping. We can see that by increasing the concentration of nanoparticles, the values of the parallel and perpendicular components of dielectric tensor tended to increase; however, the difference between them decreased as the orientation became less uniform. As shown here, this was not the case for the 5CB + 1%AzoGNPs cell elastic constant, which increased by 5.5%. We hypothesize that the aggregation process did not occur for this concentration, which agrees with the absence of particle precipitation at the bottom of a vial.

A similar behaviour in dielectric anisotropy and elastic constants was reported in the literature. Vardanyan et al. reported different doping concentrations of GNPs in 5CB [33,42]. They showed that both elastic constants and the dielectric anisotropy tend to decrease above a critical concentration, where gold aggregates start to form. They also discussed the increase in elastic constants, when the doping content is below this critical value, relating it to the formation of LC–gold conglomerates. The influence of GNPs content on the electrical, elastic and rheological properties of a nematic liquid crystal has also been studied by Chausov et al. [16]. Using gold nanoparticles of size between 7 and 15 nm, they doped ZhK-1289 LC up to 5 wt%. They reported an increase in the splay elastic constant and rotational viscosity at a low doping concentration (<1 wt%) followed by a decrease in these parameters as the GNPs concentration increased. This trend is similar to what we observed in 5CB, with K_1_ and γ_1_ increasing in the AzoGNPs suspension with the doping content of 1 wt% and then decreasing in the suspension with 3 wt% AzoGNPs concentration. However, both these groups reported an increase in Δε at low concentrations of GNPs, which is not in agreement with our study of AzoGNPs.

In the case of ferroelectric nanoparticles, Δε increased 2.7 times while K_1_ decreased by 20%. Compared to the small GNPs (5 nm core), SPS nanoparticles are much larger (50 nm) and thus cause a stronger distortion of the LCs around them even if they are well dispersed, leading to a decrease in K_1_. In contrast, the huge increase in dielectric anisotropy can be explained by the polarizability of ferroelectric inclusions, as reported before [14,43,44].

### 3.2. Rotational Viscosity Measurements

Another key parameter of LCs is rotational viscosity γ_1_, which determines the response time of LC devices. The viscosity of a liquid crystal can be significantly influenced by doping with nanoparticles. It has been reported that the viscosity of such colloidal system tends to increase for small amounts of dopants and to decrease when nanoparticles start to aggregate [33,42].

The rotational viscosity was measured using OMPA as described in Section 2.3. The experimental data were fitted with the OMPA algorithm using rotational viscosity as a fitting parameter [39]. The experimental data and fits for pure and doped 5CB are shown in Figure 3, and values for rotational viscosity for all suspension are given in Table 1. We can observe from Figure 3 that the fit and the experimental data match the slope part of the CPI very well, which was then used by the model to estimate the rotational viscosity γ_1_ [37]. This, once again, demonstrates that the OMPA model can fit nanoparticle-doped LCs.

The rotational viscosity increased from 92.5 mPa.s for undoped to 98.41 mPa.s for 5CB + 1%AzoGNPs. However, with the further increase in AzoGNPs content in the suspension to 3%, the rotational viscosity dropped to 89.03 mPa.s. This observation could be explained by the formation of irregular-shaped aggregates that disturb the LCs in the surroundings, as the concentration of AzoGNPs increases. The presence of gold nanoparticle aggregates tends to lower the viscosity of the suspension as the liquid crystal has more space to move freely, whereas well-dispersed GNPs may interfere with the reorientation process [6,42].

The rotational viscosity fit for E7 + 1 wt% AzoGNPs can be found in the Appendix A, also showing a good agreement with the experimental data. For this suspension, we observed that the rotational viscosity decreased from 203 mPa.s for the undoped E7 to 148.17 mPa.s in the suspension with 1% of nanoparticle content. Such a trend indicates the presence of particle aggregates in the suspension. No studies were carried out on E7 + 3%AzoGNPs as it was not possible to obtain a stable, colloidal suspension at this concentration.

For the LC18523 + SPS suspension, we obtained an increase in the viscosity by a factor of 1.4 in comparison with the undoped LC18523. In contrast to AzoGNPs, the interaction between SPS NPs and LC molecules is mainly due to the strong dipole moment of the nanoparticles. This strong interaction induces local areas where LC molecules are attracted to the nanoparticles, and thus are more difficult to reorient, increasing the average viscosity of the suspension.

This suspension was studied by us earlier in [35], where we estimated the rotation viscosity by measuring a response time of a twist cell. The previous results suggest that doping with 1 wt% of SPS increased the rotational viscosity by a factor of 3.2. Hence, we suggest that some degree of particle aggregation has occurred over time. As discussed earlier, the increase in the degree of particle aggregation can lead to the decrease in the rotational viscosity in the suspension from a factor of 3.2 [35] to a lower value, as seen in the current study, in which our repeated measurements showed a value of 1.4. However, we can still observe that the rotational viscosity of the doped LC18523 remains higher than that of the undoped version of this liquid crystal, meaning that the aggregation process is not solely responsible for the increased viscosity observed.

### 3.3. Elastic Constants and Stability of Ferroelectric Nanoparticle-Doped LCs

In this section we further analyse the stability of the SPS colloidal suspension. To do so, we compared OMPA fits obtained using experimental data measured previously for LC18523 and LC18523 + 1 wt% SPS suspensions [35], with new measurements of the same suspension. The CPI traces in [35] were measured using a laser beam at 633 nm, and the new experiments were performed using a wavelength of 532 nm. Fitting the previous data at 633 nm allowed us to compare our results with the values from the literature [35] and ensure the precision and capacity of the OMPA model in characterising such colloidal LC systems [45,46]. By repeating this measurement using the same samples, we checked the stability of the suspension over time. The results are summarised in Table 2 and shown on Figure 4.

From Figure 4 and Table 2, we can notice that OMPA provides a good-quality fit to the previously reported data at 633 nm, and the values of the elastic constants show good agreement with the literature values [35]. However, we could not obtain the same good-quality fit for the repeated experiments at 532 nm. This is also reflected in some deviation in the fitted values of the elastic constants in comparison to the elastic constants obtained at 633 nm. The splay elastic constant K_1_ decreased from 7.1 pN to 6.5 pN, while the bend elastic constant K_3_ increased from 10.5 pN to 11 pN. This observation was reproducible across the cell.

The decrease in K_1_ can be explained by the quality of the experimental CPI trace at low voltage, where the first two maximums had a reduced value (see Figure 4b), leading to an underestimation of K_1_ during fitting [40]. As the samples were old, we believe that this is another sign that some degree of aggregation occurred in the suspension. The aggregates caused nonuniform liquid crystal orientation around them. This effect became especially prominent at voltages around the threshold value, when LC orientation was not uniform across the cell. This nonuniformity in LC alignment resulted in an increase in light scattering at this voltage range, observed by the decrease in the height of the first CPI maximum. The increased value of K_3_ could be due to the lack of agreement between fitting and experimental curves at a high-voltage range, the part of CPI trace that is used to extract K_3_ [40].

As the model from the OMPA shows a very good agreement at 633 nm for both the fitted and literature values of K_1_ and K_3_, we can conclude that the OMPA parameters are reliable. The difference in the magnitude of the elastic constants for doped LC18523, from the current fitting and the previously published work, confirms the trend observed for the long-term rotational viscosity changes, suggesting that some degree of aggregation of nanoparticles is the underlying mechanism.

## 4. Conclusions

We have demonstrated that OMPA can successfully characterise nematic LCs doped with different nanoparticles, such as ferroelectric and gold nanoparticles. In particular, we prepared, and then characterised for the first time to our knowledge, a stable suspension of AzoGNPs in E7. Knowing only dielectric coefficients and refractive indices, we were able to extract both elastic constants and the rotational viscosity of pure and doped LCs, the parameters that are important for the electric field and time response. This result is, indeed, consistent with homogenisation studies of liquid crystal suspensions, which show that the suspension can be modelled by an effective theory, in which the LC crystal parameters are altered by the presence of the nanoparticles. The values of elastic constants and rotational viscosity were also used to capture the long-term stability and ageing changes in the cells, as in the case investigated here, identifying the process of aggregation in colloidal suspensions of ferroelectric nanoparticles, such as LC18523 doped with SPS nanoparticles. Therefore, we showed that OMPA is a very versatile and useful tool able to characterise not only nematic single components and mixtures but also doped LC systems.

## Figures and Tables

**Figure 1 nanomaterials-12-00341-f001:**
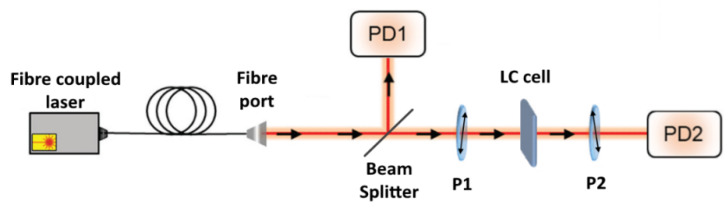
Experimental OMPA setup. A fibre-coupled diode laser beam is split so a portion of the light is directed to a first photodiode PD1 (reference signal), and the other part passes through the first polariser P1, the LC cell, and the second polariser P2 (analyser) before reaching the second photodiode PD2. The polariser and analyser axes are perpendicular to each other. This entire setup is enclosed to avoid any background noise.

**Figure 2 nanomaterials-12-00341-f002:**
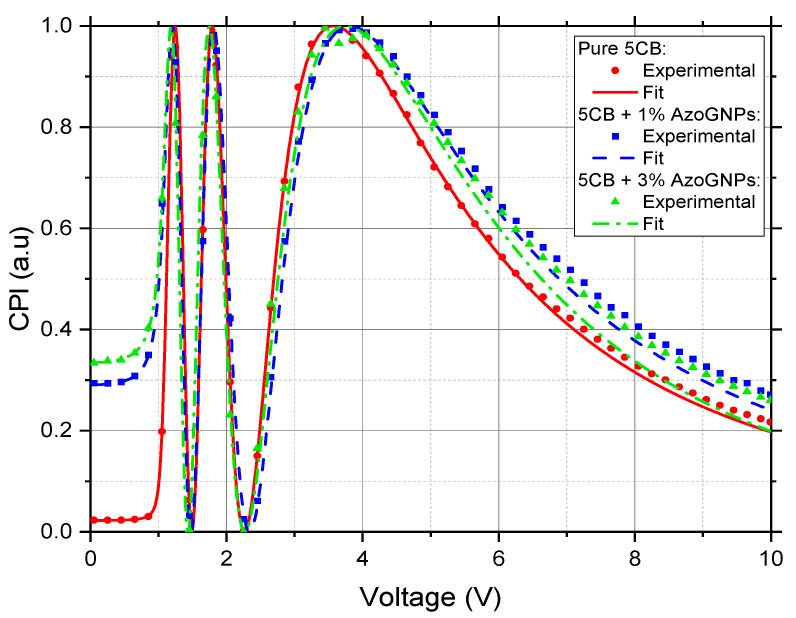
CPI measurements of pure and AzoGNP-doped 5CB under light illumination at 642 nm and a frequency of an applied electric field of 10 kHz. Experimental data are represented by symbols, the OMPA fits by lines.

**Figure 3 nanomaterials-12-00341-f003:**
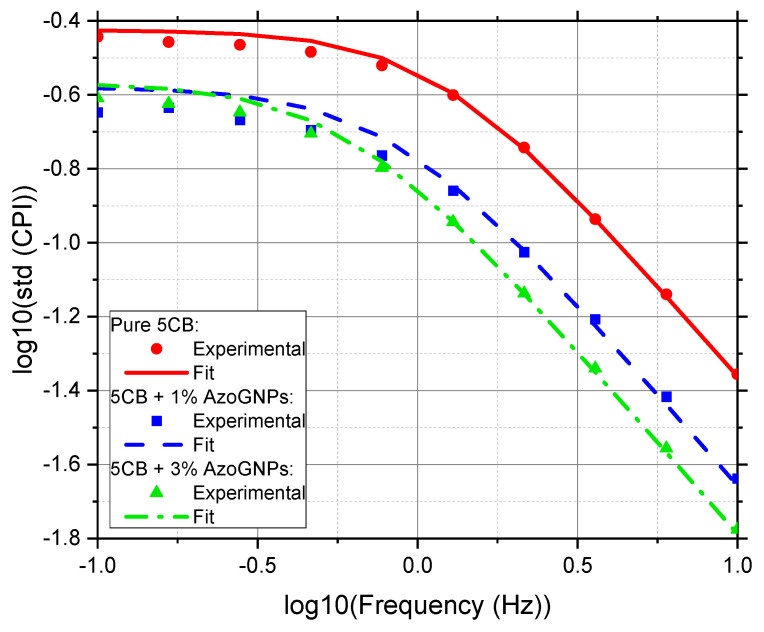
Rotational viscosity measurements of pure and nanoparticle-doped 5CB under light illumination at 642 nm. Experimental data are represented by symbols, and the OMPA fits are shown by solid lines.

**Figure 4 nanomaterials-12-00341-f004:**
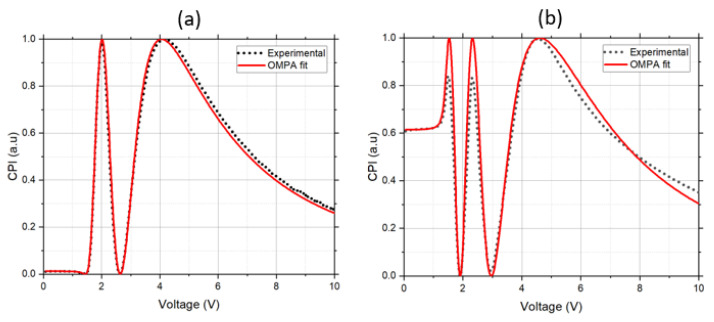
Experimental CPI measurements (black dots) and corresponding OMPA fit (solid red line) of LC18523 + 1 wt% SPS at 633 nm (**a**) and 532 nm (**b**) at 10 kHz. The CPI data fitted at 633 nm (**a**) was collected and published previously [35], and the CPI data fitted at 532 nm (**b**) were measured recently. The fitted parameters that correspond to these fits are presented in Table 2.

**Table 1 nanomaterials-12-00341-t001:** Elastic constants and rotational viscosity of pure and nanoparticle-doped E7, 5CB and LC18523 obtained through OMPA fitting. The changes of the dielectric anisotropy through doping are also shown: Δε decreased with the addition of AzoGNPs but increased with ferroelectric nanoparticles. The errors on the OMPA fits were studied previously for elastic constants [40] and are estimated to be approximately 1% for K_1_ and 2% for K_3_. The same has been carried out for viscosity measurements [38], and the error was estimated in the range of 1%.

	λ (nm)	K1 (pN)	K3 (pN)	ε_//_	ε_⊥_	Δε	γ_1_ (mPa.s)
5CB	642	6.0	9.0	17.9	6.5	11.4	92.50
5CB + 1%AzoGNPs	642	6.3	9.3	21.8	11.4	10.4	98.41
5CB + 3%AzoGNPs	642	4.1	7.5	24.3	17.5	6.8	89.03
E7	642	10.7	16.2	19.5	5.2	14.4	203
E7 + 1%AzoGNP_S_	642	8.9	16.6	17.6	7.5	10.1	148.17
LC18523	532	8.2	9.1	6.7	4.2	2.5	92.7
LC18523 + 1%SPS	532	6.5	11.0	11.5	4.7	6.8	128.58

**Table 2 nanomaterials-12-00341-t002:** Summary of elastic constants obtained by the OMPA through the fitting of CPIs from cells filled with pure and nanoparticle-doped LC18523. These cells were measured using light illumination at 633 nm and 532 nm.

	λ (nm)	K1 (pN)	K3 (pN)
Pure LC18523 (literature [35])	633	7.85	10.0
Pure LC18523 (OMPA fit of [35])	633	7.9	10.1
Pure LC18523 (OMPA new measurements)	532	8.2	9.1
LC18523 + 1 wt% SPS (literature [35])	633	7.1	10.5
LC18523 + 1 wt% SPS (OMPA fit of [35])	633	7.3	10.5
LC18523 + 1 wt% SPS (OMPA new measurements)	532	6.5	11.0

## Data Availability

The data sets generated and analysed during the current study are available from the corresponding author on reasonable request.

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
