# Peer review of "Nanoparticle-Induced Property Changes in Nematic Liquid Crystals"

_nanomaterials, 2022, doi:10.3390/nano12030341_

Round 1
Reviewer 1 Report
The paper demonstrates the feasibility of the apparatus developed by the authors some years ago to measure several physical properties of nematic liquid crystal doped by nanoparticles. It is clear, interesting and deserves publication.
A small improvement in the presentation should be done to answer the following questions:
- what is the preliminary cell thickness of the sample in the preparation ? how is it fixed ? (it is clear that the actual thickness is determined by data fitting)
- the authors mention that refractive index of materials has been determined in previous work. However it seems that previous work has been done only in LC doped by ferroelectric nanoparticles, what about the samples doped by gold nanoparticles ?
Author Response
Many thanks for the time you spent reviewing our manuscript. Here are our answers to your questions. We added some details in the revised manuscript, these are highlighted by the change tracker mode from Word, and the corresponding lines are listed in the answers below.
- What is the preliminary cell thickness of the sample in the preparation ? How is it fixed ? (it is clear that the actual thickness is determined by data fitting)
Cells with LC18523 and LC18523+SPS were prepared 10 years ago using 12µm spacer to fix the LC thickness. Suspensions of GNPs in E7 and 5CB were tested in commercial planar cells with a set thickness of 10±0.5µm. Details have been added in the revised manuscript. (line 111-118)
- The authors mention that refractive indices of materials have been determined in previous work, what about the samples doped by gold nanoparticles ?
Unlike the doped LC18523, we did not observe any significant changes in birefringence from doped or undoped 5CB/E7 using CPI measurements, as evidenced by figure 2.
We are preparing a separate paper on investigating errors of LC parameters on fitting quality, which shows that the actual values of no and ne are not critical parameters if birefringence is known. Therefore, we used the values of the undoped 5CB/E7 when fitting the suspensions.
Reviewer 2 Report
December 31, 2021
Comments on the paper “Quick method for measuring nanoparticles induced property changes in nematic liquid crystals.”
The authors have carried out elastic constant and viscosity measurements on composites of nematic liquid crystals with gold/ ferroelectric nanoparticles by employing what they claim as an optical multiparameter analyzer. The technique is not new and has already been employed in a few of their recent publications. Therefore the title “Quick method for measuring ……………” is not justifiable, notwithstanding the feature that in the current work, nanoparticles are employed. Even the calculations are not new. Besides, the observations that mixtures with a particular concentration of nanoparticles work alright for 5CB, but not for the structurally similar E7 demand clear explanations.
The behaviour of different parameters should be compared with those in the literature.
The article is unacceptable in the current format and requires major restructuring.
Minor point
- Authors should cite some of the recent references viz., Adv. Opt Mater 7, 1801408 (2019); J. Mol Liq (2021; doi.org/10.1016/jmolliq.2021.118004, Technologies 7, 32 (2019); J. Phys. D: Appl. Phys. 55 083002 (2022)
Author Response
Many thanks for the time you spent reviewing our manuscript. Here are our answers to your questions. We added some details in the revised manuscript, these are highlighted by the change tracker mode from Word, and the corresponding lines are listed in the answers below.
- The title “Quick method for measuring ……………” is not justifiable, notwithstanding the feature that in the current work, nanoparticles are employed
The referee is correct that the title might be confusing. The paper is about the impact of nanoparticles in nematic LCs and the suspension time stability measurements. The OMPA is the perfect tools to check it quickly and easily.
Even if we have shown in the paper that the method also works for doped LCs, the core of the paper is about showing the impact of FNPs and GNPs rather than describing a method. Therefore, we changed the title to “Nanoparticles induced property changes in nematic liquid crystals”.
- The title “Quick method for measuring ……………” is not justifiable, notwithstanding the feature that in the current work, nanoparticles are employed
The referee is correct that the title might be confusing. The paper is about the impact of nanoparticles in nematic LCs and the suspension time stability measurements. The OMPA is the perfect tools to check it quickly and easily.
Even if we have shown in the paper that the method also works for doped LCs, the core of the paper is about showing the impact of FNPs and GNPs rather than describing a method. Therefore, we changed the title to “Nanoparticles induced property changes in nematic liquid crystals”.
- Mixtures with a particular concentration of nanoparticles work alright for 5CB, but not for the structurally similar E7 demand clear explanations
Azo-GNP have so far only been studied in 5CB, a single component liquid crystal. In this paper we have expanded the literature to include also E7, a mixture of several mesogenic molecules. The solubility of nanoparticles in a liquid crystal depends strongly on the interaction between ligands and liquid crystal molecules. We believe that this structural difference may be the cause of the change in maximum concentration of azo-GNP in the two liquid crystals. However, this require a more in-depth investigation, which is outside of the scope of this paper. We added a sentence to tell the readers that 5CB and E7 have different structures. (line 173-174)
- The behaviour of different parameters should be compared with those in the literature
The general behaviour of Azo-GNPs doped E7 and 5CB has been compared with other type GNPs used in the literature. We used this to relate the evolution of the splay elastic constants, dielectric anisotropy and viscosity to aggregation as describe in the literature [31,38]. We have added in the text, more explicitly, the effect of GNPs reported before and the commented on the agreement with our results. (line 215-223)
- Authors should cite some of the recent references viz., Adv. Opt Mater 7, 1801408 (2019); J. Mol Liq (2021; doi.org/10.1016/jmolliq.2021.118004, Technologies 7, 32 (2019); J. Phys. D: Appl. Phys. 55 083002 (2022)
We thank you for suggesting the references, we added three of them in the introduction. Unfortunately, we could not find the second reference as the information is incomplete.
Reviewer 3 Report
My report is uploaded as a PDF file.

Author Response
Many thanks for the time you spent reviewing our manuscript. Here are our answers to your questions. We added some details in the revised manuscript, these are highlighted by the change tracker mode from Word, and the corresponding lines are listed in the answers below.
- What is the area of the sample that is measured (probably the area illuminated by the laser beam)?
The region measured correspond to the area illuminated by the laser, which is around 7 mm2. This information has been added to the revised manuscript. (line 129-132)
- Did the authors try to repeat their measurements in several parts / areas of the cell? This could probe parts without and with (or with higher and lower degree of) aggregation and demonstrate the differences. Or maybe this has been already done and an averaging over the signal at different areas is used for estimating the output parameters? Please add the information or some comments on that in the revision.
We have taken multiple measurements in the LC18523+SPS cell and did not notice any significant differences between them. We could not repeat this procedure using the commercial cells, in view of the limited electrode area (5x10 mm) with respect to the beam diameter (approximately 3 mm). However, visual inspection of these cells suggested that also for these cells there was no significant inhomogeneity. (line 121-124 and 297)
- Apart from 5CB which is extensively studied and known liquid crystal compound (being nematic at room temperature), there are measurements of E7 and LC18523. The authors may add the information about the phase of these liquid crystals at the temperature where the measurements were performed (probably nematic?).
Both E7 and LC18523 are nematic at room temperature, we added the information in the cell preparation section. (line 124-125)
- Regarding the measurement of aged sample (10 years) this is very interesting, since the stability over long time scales is important for potential applications. Some crucial information is missing here. For example, were the samples kept over this time at constant (room?) temperature and phase (nematic or crystal)? Were the samples heated again (or remixed / sonicated) prior to the new measurements? A comparison of an old sample without and with mixing/heating to isotropic phase would provide very valuable information. I would encourage the authors to add this if possible and comment on the above in the revised manuscript.
The LC18523 cells were kept at room temperature in ambient air (the cells were sealed). None of these cells were sonicated or heated again before new measurements. We have added this specification in the revised manuscript (line 114-116). To avoid any confusion, we want to point out the fact that we kept and study a cell filled with SPS suspension, and not a suspension flask that we used again to fill a new cell. This does not allow us any mixing or sonication of any kind.
- It would be reasonable to add some error bars in the estimated quantities in Tables 1 and 2, or at least refer to some rough estimate of error in the determination of values (in the Table captions or in the manuscript text).
These errors can only be gathered from averaging and multiple measurements. We estimated the average error from the fitting to be in the order of 1% for K1 and 2% for K3. The error on the viscosity should be in the same range as the calculation depends on the values of K1 and K3. (line 196-198)
- A kind suggestion to the authors to consider for future work: it would be great to perform similar measurements in liquid crystal chiral phases, such as cholesteric, blue phase and twist grain boundary phases with dispersed nanoparticles. In particular, there is a lot of discussion about the mechanisms behind the inclusions’ assembly in the defect lattices of frustrated chiral structures (blue phases and twist grain boundary phases). Nanoparticle stabilization of these phases is recently attracting enormous interest [see for example: Draude et al., Nanoscale Advances 2: 2404 (2020), Yadav & Malik, Optical Materials 122: 111670 (2021]. Apart from the stabilization effect [see for example: Cordoyiannis et al., Nanomaterials 11: 2968 (2021)] there are reports of novel plasmonic properties [see for example: Shi et al., New Journal of Chemistry 39: 1899 (2015), Sahoo et al., Journal of Molecular Liquids 299: 112117 (2020)] or improved electro-optical properties due to the presence of nanoparticles [see for example: Liu et al., Physical Review E 89: 052505 (2014)].
We want to thank you for these useful suggestions and references. For now, the OMPA model only work for planar cell. We currently are working on a version to work with twisted cell too, but the model will have to be adapted to work with the suggested LCs.
Round 2
Reviewer 2 Report
January 13, 2022
Comments on the paper “Quick method for measuring nanoparticles induced property changes in nematic liquid crystals.”
The authors have modified the manuscript, but it still requires revision/corrections. After incorporating the suggested discussion, modification, corrections, the manuscript may be accepted for publication.
- The authors should compare bring in discussion with respect to papers such as J. Phys.: Condens. Matter 32 (2020) 395102, etc.,
- The title should be “Nanoparticle induced….” not “Nanoparticles induced….”.
- However, the limited size of the electrode for commercial cells compared to our beam diameter did not allow us to probe the cells and take multiple measurements.
It is not clear, do the authors mean measurement couldn’t be done at different points of the sample or what?
- “…where the LCs and suspensions are in the nematic phase…..”
Once it is a uniform mixture, it is a single component. Why the separate mention of “LCs and suspension”
- The reference is correct. Mention here is the DOI again https://doi.org/10.1016/j.molliq.2021.118004
- There are grammatical mistakes in the rewritten manuscript, has to be corrected.

Author Response
- The authors should compare bring in discussion with respect to papers such as J. Phys.: Condens. Matter 32 (2020) 395102, etc.
Thank you for the suggested reference. We have extended the section that compare our results with the literature, showing a good agreement with published data on similar topic using unfunctionalized gold nanoparticles as compared to our AzoGNPs
- The title should be “Nanoparticle induced….” not “Nanoparticles induced….”.
The mistake is now corrected in the revised manuscript.
- “However, the limited size of the electrode for commercial cells compared to our beam diameter did not allow us to probe the cells and take multiple measurements.” It is not clear, do the authors mean measurement couldn’t be done at different points of the sample or what?
We modified the sentence, so it sounds less confusing: “The OMPA set-up, in principle, can map a large area of a cell, determined by the size of the illuminating beam and the size of the ITO region [43]. However, in case of commercial cells, their limited size of the electrode did not allow us to probe the cells at different points across the cells.” (line 162-165). Indeed, the size of the laser beam has close dimensions compared to the electrodes area, making it difficult to measure several points. To avoid having the beam partly illuminating outside of the electrode areas, we targeted the centres so we can ensure good quality data.
- “…where the LCs and suspensions are in the nematic phase...” Once it is a uniform mixture, it is a single component. Why the separate mention of “LCs and suspension”
As doping can influence the phase transition temperatures, we wanted to specify that both the doped and the undoped LCs are in the nematic phase at room temperature, the temperature at which the measurements were performed and the cells with doped and undoped LC18523 were stored. We change the terms “LCs and suspensions” to “undoped and doped LCs” (line 125).
- The reference is correct. Mention here is the DOI again https://doi.org/10.1016/j.molliq.2021.118004
We added the suggested paper as reference [27] in the introduction.
- There are grammatical mistakes in the rewritten manuscript, has to be corrected.
We revised the whole text of the manuscript and corrected a couple of mistakes that escaped our attention earlier. We also took this opportunity to revise the style to make our arguments clearer. All the changes are visible under track changes and none of them changed the substance of our arguments, just improve their presentation.